# Relationship between Aggressiveness, Self-Confidence, and Perceived Coach Support and Head Impact Exposure in Youth Football

**DOI:** 10.3390/sports10080115

**Published:** 2022-07-29

**Authors:** Madison E. Marks, William C. Flood, Mireille E. Kelley, Mark A. Espeland, Christopher M. Miles, Alexander K. Powers, Christopher T. Whitlow, Joseph A. Maldjian, Joel D. Stitzel, Jillian E. Urban

**Affiliations:** 1Department of Biomedical Engineering, Wake Forest University School of Medicine, Winston-Salem, NC 27101, USA; memarks@wakehealth.edu (M.E.M.); mireille.kelley1@gmail.com (M.E.K.); apowers@wakehealth.edu (A.K.P.); cwhitlow@wakehealth.edu (C.T.W.); jstitzel@wakehealth.edu (J.D.S.); 2Virginia Tech-Wake Forest University School of Biomedical Engineering and Sciences, Winston-Salem, NC 27101, USA; 3Department of Neuroscience, Wake Forest School of Medicine, Winston-Salem, NC 27101, USA; wflood@wakehealth.edu; 4Department of Radiology, Wake Forest School of Medicine, Winston-Salem, NC 27101, USA; 5Department of Gerontology and Geriatric Medicine, Wake Forest School of Medicine, Winston-Salem, NC 27101, USA; mespelan@wakehealth.edu; 6Department of Family and Community Medicine, Wake Forest School of Medicine, Winston-Salem, NC 27101, USA; cmmiles@wakehealth.edu; 7Department of Radiology, University of Texas Southwestern Medical Center, Dallas, TX 75390, USA; joseph.maldjian@utsouthwestern.edu

**Keywords:** biomechanics, head acceleration, self-confidence, coach support, aggression

## Abstract

This study evaluated head impact exposure (HIE) metrics in relation to individual-level determinants of HIE. Youth (n = 13) and high school (n = 21) football players were instrumented with the Head Impact Telemetry (HIT) system during one season. Players completed the Trait-Robustness of Self-Confidence Inventory (TROSCI), Sports Climate Questionnaire (SCQ), and Competitive Aggressiveness and Anger Scale (CAAS), measuring self-confidence, perceived coach support, and competitive aggressiveness, respectively. Relationships between HIE metrics (number of impacts, median and 95th percentile accelerations, and risk-weighted exposure (RWE)) and survey scores were evaluated using linear regression analysis. For middle school athletes, TROSCI scores were significantly negatively associated with the number of competition impacts and the mean number of impacts per player per competition. SCQ scores were significantly positively associated with median linear acceleration during practice. CAAS scores were not significantly associated with biomechanical metrics at either level of play. Perceived coach support and self-confidence might influence HIE among middle school football players. Football athletes’ competitive aggressiveness may have less influence their HIE than other factors.

## 1. Introduction

Concussions continue to be a prevalent safety concern in contact and collision sports due to their potential short- and long-term neurodegenerative effects. Studies suggest that repetitive, non-concussive head impact exposure (HIE) may lead to similar pathologies as those caused by concussion [1,2,3,4,5,6]. With approximately 5 million athletes between the ages of 6 and 18 participating in football each year, it is essential to minimize HIE to prevent future brain injury [7,8,9,10].

Previous studies have shown that team-based activities, such as practice drills and game play, affect HIE of athletes [10,11,12,13,14]. In an attempt to reduce head impacts among athletes, youth football organizations have begun to restrict the amount of contact teams can engage in during practice, and some require coaches to receive certification/training on safe tackling [15,16]; while this has been shown to reduce HIE overall, HIE still varies greatly among individual athletes [11,12,16,17,18,19,20,21,22]. The differences in HIE between individuals might be affected by personal characteristics and behaviors [21,22].

Competitive aggression, perceived coach support, and self-confidence may play a role in the HIE that an athlete experiences. Aggression has been identified as a potential influencing factor in hockey HIE and has been associated with higher rotational acceleration at practices in a cohort of youth hockey players [19]. It is hypothesized that coach interactions and communication with athletes might influence HIE in coach-directed practice drills and gameplay, as some studies have shown that coaches may affect other behaviors surrounding head impacts (e.g., athlete concussion reporting) [11,12,23,24,25,26,27]. Lastly, self-confidence has long been thought to influence athlete performance, with some studies finding correlations between sport confidence and subjective performance [28,29,30]. Self-confidence might affect HIE through an athlete’s confidence in technique and willingness to engage in or avoid collisions. However, there is limited information on how individual-level determinants (i.e., characteristics internal to the individual) affect an athlete’s HIE [19,31]. Therefore, the objective of this study was to examine the effect of confidence in sports, perceived coach support, and competitive aggressiveness and anger in relation to HIE in football.

## 2. Materials and Methods

Demographic, survey, and head impact data were collected from youth football athletes participating on one middle school team and one high school team. This study was approved by the Wake Forest School of Medicine Institutional Review Board (Approval: IRB00014350). Written assent was obtained from participants and consent was obtained from parents. Study participation was voluntary.

Sample size was determined from a convenience sample of athletes enrolled in the parent studies. Only athletes with complete sets of biomechanical and survey data were included in this study. The biomechanical data from this study will be available via FITBIR. Methodologies have been thoroughly described with references to supporting literature where necessary.

Athletes participating on one middle school-level (ages 12–14) team and one high school-level (varsity; ages 14–18) team were enrolled in this study. Participants were fitted with Riddell Speed or SpeedFlex helmets equipped with the Head Impact Telemetry (HIT) System. The HIT System utilizes spring-mounted sensors to measure frequency and acceleration of head impacts [13,20]. HIE data were collected, using the HIT System, from the participating athletes over the span of one season. Video was recorded, by research assistants, to cross-reference and validate head impact events. Impacts captured when helmets were not being worn and outside of practice and competition periods were excluded and impacts over 40 g were individually video verified. The data collection and processing methodologies have been previously described [13,20,32,33,34].

Each athlete completed the Trait-Robustness of Self-Confidence Inventory (TROSCI) [35], the Sports Climate Questionnaire (SCQ) [36], and the Competitive Aggressiveness and Anger Scale (CAAS) [37], once during this study. TROSCI items seven and eight were slightly modified to include appropriate language for the participating athletes to ensure comprehension. The SCQ and CAAS were not modified. The surveys utilized are provided in Appendix A. Middle school athletes completed the surveys at mid-season (September) and high school athletes the surveys post-season (October). The TROSCI survey includes 8 items, which participants ranked on a Likert-type scale with a value of 1 indicating strong disagreement and a value of 9 indicating strong agreement, to measure confidence. To score TROSCI, the item values were summed, after reversing the scores of items 1, 2, and 7, with a maximum possible survey score of 72 [35,38]. The higher the TROSCI score, the higher a player’s self-confidence [35,38]. The SCQ survey includes 15 items, which participants ranked on a Likert-type scale with a value of 1 indicating strong disagreement and a value of 7 indicating strong agreement, to measure perceived coach support of autonomy. To score SCQ, the item values were averaged after reversing the score of item 13, with a maximum possible SCQ score of 7 [36]. The higher a SCQ score, the more support a player perceives that they are receiving from their coach [36]. The CAAS survey includes 12 items, which participants ranked on a Likert-type scale with a value of 1 indicating almost never and a value of 5 indicating almost always, to measure anger and aggressiveness. To score CAAS, the item values were summed, with a maximum possible CAAS score of 60 [37]. The higher a CAAS score, the more aggressive the player [37]. Collinearity of survey scores within each level of play (middle school and high school) was evaluated using linear regression analysis.

HIE was quantified in terms of the total number of impacts (N), mean number of impacts per athlete per session, 95th percentile linear and rotational acceleration (LA95, RA95), and median linear and rotational acceleration (median LA, median RA). Risk Weighted Exposure (RWE), a cumulative exposure metric encompassing frequency and magnitude of impacts, was calculated using the youth concussion risk function developed by Campolettano et al. [17,39,40]. Each HIE metric was evaluated for the entire season and separately by session type (i.e., practices, competitions). Statistical analysis was completed using SAS statistical software. Data were stratified by level of play (middle school and high school). The survey scores (i.e., TROSCI, SCQ, CAAS) were compared against the biomechanical metrics for each sample using linear regression analysis, with covariates of age and body mass index (BMI), to describe the relationship between survey scores and biomechanical data. Cook’s distance (4/n) was computed to remove outliers from each regression. When comparing the survey scores and biomechanical metrics of the high school and middle school samples, the Wilcoxon sum rank test was performed due to the relatively small sample size (n < 50). Because the goals of the analysis were descriptive, no adjustment was made for multiple comparisons.

## 3. Results

Thirteen (n = 13) middle school-level and 21 high school-level football players were instrumented with HIT system for the full season and completed the surveys. A description of participants by sample is shown in Table 1. Summary statistics for each survey score for the samples are provided in Table 2.

The median scores were similar across levels of play for the TROSCI. The high school players scored higher on the SCQ and CAAS than the middle school level players, but the differences were not significant. There were no significant differences in survey scores between athletes of varying racial groups. Summary statistics for biomechanical metrics are displayed in Table 3 and Table 4 (impact frequency and impact magnitude).

When evaluating the differences in HIE, the middle school athletes had significantly higher 95th percentile linear acceleration during practices (*p* = 0.014) than their high school counterparts and the high school athletes had significantly higher median and 95th percentile rotational acceleration during competition (*p* = 0.002, *p* = 0.003) than the middle school athletes. High school athletes also had higher overall median rotational acceleration (*p* = 0.020) and higher median linear acceleration during competition (*p* = 0.011).

When evaluating survey scores regressed to each other, TROSCI scores were significantly positively associated with SCQ scores at the high school-level (*p* = 0.001, R^2^ = 0.697), with age and BMI having significant effects (negative; positive). There were no significant associations between surveys at the middle school-level; however, for the linear regressions between TROSCI scores and CAAS scores and between SCQ scores and CAAS scores, age and BMI had significant positive effects for the middle school athletes (all *p* < 0.05).

TROSCI scores were significantly associated with some biomechanical metrics among the middle school athletes, but these trends were not observed among the high school sample. Among the middle school athletes, the TROSCI score was significantly negatively associated with the number of competition impacts (*p* = 0.045, R^2^ = 0.508) and the mean number of impacts per athlete per session during competition (*p* = 0.045, R^2^ = 0.508). The TROSCI scores for the subset of high school players were not significantly associated with any biomechanical metric. For middle school athletes, BMI was significantly positively associated with median rotational acceleration during practice. Among the high school athletes, age had significant positive associations with the number of impacts (overall, practice, competition) and the mean number of impacts per athlete per session (competition). BMI was significantly positively associated with the number of impacts per player per session (practice) and significantly negatively associated with 95th percentile linear acceleration (overall, competition), 95th percentile rotational acceleration (overall, practice, competition), and RWE (practice) for the high school athletes. The strongest correlations for the TROSCI analyses in middle school and high school athletes are shown in Figure 1.

SCQ scores were significantly associated with some biomechanical metrics among the middle school athletes, but these trends were not observed among the high school sample. The SCQ scores for the middle school sample were significantly positively associated with the median linear acceleration (*p* = 0.025, R^2^ = 0.670) during practice. The SCQ scores for the high school sample were not significantly associated with any biomechanical metric. For middle school athletes, age was significantly positively associated with RWE during competition and BMI was significantly positively associated with median linear and rotational accelerations (practice; overall, practice). Among the high school athletes, age had significant positive associations with the number of impacts (overall, competition), the mean number of impacts per player per session (competition), and the median and 95th percentile rotational accelerations (overall). BMI had significant negative associations with 95th percentile linear acceleration (overall, competition), 95th percentile rotational accelerations (overall, practice, competition), and RWE (overall, practice). The strongest correlations for the SCQ analyses in middle school and high school athletes are shown in Figure 2.

CAAS scores were not significantly associated with any biomechanical metric for middle school or high school athletes. For middle school athletes, age and BMI were not significantly associated with any biomechanical metric. Among the high school athletes, age had significant positive associations with the number of impacts (overall, competition), the mean number of impacts per athlete per session (competition), and median rotational acceleration overall. BMI was significantly negatively associated with 95th percentile linear and rotational accelerations (overall; overall, practice) and median rotational acceleration (overall). The strongest correlations for the CAAS analyses in middle school and high school athletes are shown in Figure 3. All significant values for the middle school and high school samples are noted in Table 5 and Table 6, respectively.

## 4. Discussion

This study examined the effect of confidence in sports, perceived coach support, and competitive aggressiveness and anger in relation to HIE metrics in youth football. Trends were observed among all survey measures and many biomechanical metrics. Significant relationships were observed between biomechanical metrics and TROSCI and SCQ scores at the middle school level of play; at the high school level, no survey scores were found to be significantly associated with a biomechanical metric. The surveys may have better association with HIE in middle school athletes than high school athletes, although the middle school sample size was smaller. TROSCI scores were significantly associated with SCQ scores for high school athletes; this relationship might indicate that more self-confident athletes feel like they have more coach support. For high school athletes, age was most often significantly associated with number of impacts (overall, competition) and the mean number of impacts per player per competition. BMI was most often significantly associated with 95th percentile linear and rotational accelerations (overall; overall, practice). For the middle school sample, age was most often significantly associated with CAAS scores and BMI was most often significantly associated with median rotational acceleration at practice and CAAS scores.

Normative values from other studies were examined for comparison to the aforementioned results. Beattie et al. found mean TROSCI scores of 35.5 and 38.1 (mean age: 19.2 years old), which are lower than the scores for the middle school and high school samples in this study [35]. For the SCQ, another study on collegiate athletes found an average score of 2.45 out of 7, which is lower than the mean scores for both samples in this study [41]. An initial study on CAAS found the mean scores to be 52.31 and 46.08 for males participating in contact and non-contact sports, respectively (mean ages for studies: 21.8 and 25.1 years old) [37]. These values are higher than the middle school and high school scores for this study [37]. Differences between the results of this study and past research might be due to variations in the age of study participants and sports in which athletes participated.

TROSCI scores, which represent self-confidence, were associated with two HIE metrics. TROSCI scores were negatively associated with the number of competition impacts and the mean number of impacts per player per session during competition, for the middle school sample. In contrast to the trends observed among practice impacts, these results may imply that more self-confident players might be engaging in contact during games less often, possibly by outrunning or dodging opponents, or that more confident athletes may be better at anticipating contact or applying use of proper technique to remove their head during contact, though more research is needed to understand the role of self-confidence and involvement in contact events in football [42,43].

SCQ scores, which represent perceived coach support, were associated with one HIE metric in middle school athletes. SCQ scores were positively associated with the median rotational acceleration during practice, for the middle school sample. Some middle school football teams might employ coaches who are parents or relatives of athletes, while this is less common for high school football teams. Middle school-level teams are also smaller in size, and athletes often have more direct interaction with their coaches during practice than their high school counterparts; therefore, perceived coach support might translate to coach influence over the athletes. If harder hits are celebrated by the coaching staff, this may further encourage risky behavior by athletes to impress their coaches. In the opposite way, coaches that are mindful of safe tackling and encourage this behavior in their athletes might influence athletes to engage in situations that might lead to HIE less often. Studies evaluating other sports have found that perceived support of autonomy by coaches might motivate athletes to engage more in their sport [44,45,46,47]. Significant associations were not observed in the high school sample; this may be because those athletes might experience fewer one-on-one interactions with all of their coaches and often participate in position-specific activities which have been shown to influence HIE [48].

CAAS scores, which represent competitive anger and aggressiveness, did not have associations with HIE metrics for either the high school or middle school samples. These results may indicate that player aggression has less influence on HIE in football athletes than other factors. A study on ice hockey found the opposite tendency, with more aggressive athletes being more likely to sustain higher severity head impacts than less aggressive athletes [19].

Differences in impact magnitude were observed between the samples by session type. Greater 95th percentile rotational acceleration and median linear acceleration, both during competition, among the high school sample is congruent with prior studies [9,39]. High school football players are generally bigger and stronger than middle school football players and, thus, may be able to contact with a greater force. Additionally, high school football players may have more advanced skills allowing them to better anticipate other players’ moves and tackle from oblique positions. On the contrary, greater 95th percentile linear acceleration during practices was observed among the middle school athletes. This may be due to differences in practice structure and contact limitations at the two levels of play. The middle school team involved in this study did not have time restrictions on the amount or type of contact that they could experience during practice, with some restrictions on the type of drills conducted [11,12]. High school practices involved more advanced technical drills and focused less on head-on impacts, therefore causing those athletes to have less linear acceleration at practice. The high school athletes were also restricted in the amount of time that they could spend on contact drills in practice.

The results of this study provide insight into individual determinants that might influence HIE in youth football athletes; however, limitations of this study should be considered. Multiple comparisons corrections were not accounted for in the analyses, which may affect the interpretation of the statistical results. Other determinants, not included in this study, may have an influence on the HIE that an individual athlete might experience. The sample sizes for the middle school team (n = 13) and the high school team (n = 21) are relatively small due to exclusion of participants with incomplete data sets. Each athlete’s scores, for the surveys, have the potential to change regularly across a season. Athletes completed the surveys once during this study, and it is possible that they may have had varying scores at different time points throughout the season. The middle school and high school athletes completed their surveys at different time points during the season (middle school: mid-season, high school: end of regular season); this might have led to differences in athlete attitudes towards competition and their coach at the time of survey completion. The surveys utilized were not necessarily designed for youth athletes; however, TROSCI items seven and eight were modified to better ensure comprehension and the SCQ and CAAS were expected to be coherent for middle school and high school athletes. The middle and high school teams that participated might follow different rules and coaching than other youth football teams; therefore, they might experience different HIE, and the results of this study cannot be generalizable to every team across the country. The HIT system has an individual measurement error of up to 15.7% and an average measurement error of approximately 1–3% for large groups of measurement data [13,14,32].

## 5. Conclusions

Relationships between HIE metrics and self-confidence, perceived coach support, and aggressiveness and anger were evaluated for a middle school sample and a high school sample. The results of this study indicate that the aforementioned characteristics may influence HIE among athletes and could be used to initiate discussions with athletes and coaches on self-confidence, perceived coach support, and competitive aggressiveness and anger in relation to football. Further studies will be conducted to examine possible individual determinants in youth football. Future intervention studies should consider the perceptions and characteristics of athletes and the role of coach in possibly influencing HIE.

## Figures and Tables

**Figure 1 sports-10-00115-f001:**
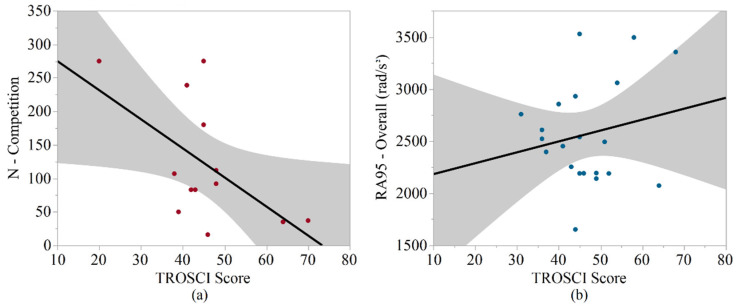
Strongest correlations between biomechanical metrics and TROSCI scores. (**a**) Linear regression for the mean number of competition impacts, for the middle school samples versus TROSCI scores. (**b**) Linear regression for the 95th percentile rotational acceleration overall, for the high school samples versus TROSCI scores.

**Figure 2 sports-10-00115-f002:**
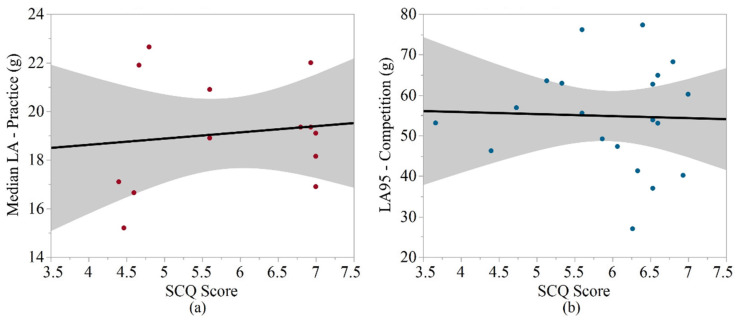
Strongest correlations between biomechanical metrics and SCQ scores. (**a**) Linear regression for the median linear acceleration (practice), for the middle school samples versus SCQ scores. (**b**) Linear regression for the 95th percentile linear acceleration (competition) for the high school samples versus SCQ scores.

**Figure 3 sports-10-00115-f003:**
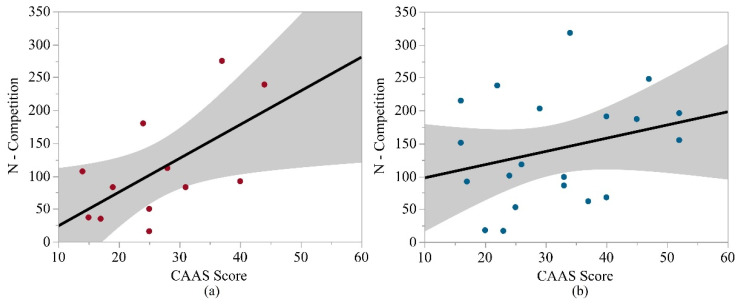
Strongest correlations between biomechanical metrics and CAAS scores. (**a**) Linear regression for the number of competition impacts, for the middle school samples versus TROSCI scores. (**b**) Linear regression for the number of competition impacts, for the high school samples versus TROSCI scores.

**Table 1 sports-10-00115-t001:** Demographics of participants by sample. (**a**) Mean and standard deviation values for age, height, and weight of participants. (**b**) Distribution of participants by race.

	**(** **a)**	
Characteristic	Middle School	High School
Age (years)	13.3 ± 0.4	16.0 ± 0.9
Height (m)	1.69 ± 0.07	1.79 ± 0.08
Weight (kg)	63.5 ± 8.1	90.0 ± 25.9
BMI (kg/m^2^)	22.3 ± 2.9	27.8 ± 6.6
**(b)**		
Race	Middle School (# of Athletes)	High School (# of Athletes)
American Indian/Alaska Native	0	0
Asian	0	0
Pacific Islander	0	0
Black/African American	9	11
White	2	7
Two or more of the above races	1	3
Declined	1	0

**Table 2 sports-10-00115-t002:** Statistical summary of survey scores for each sample *.

Survey	Statistic	Middle School	High School
TROSCI	Mean ± SD	45.3 ± 12.1	46.6 ± 9.2
	Median [95th %]	45.0 [70.0]	45.0 [64.0]
SCQ	Mean ± SD	5.83 ± 1.13	5.75 ± 1.25
	Median [95th %]	5.60 [7.00]	6.27 [6.93]
CAAS	Mean ± SD	26.7 ± 9.4	31.9 ± 11.4
	Median [95th %]	25.0 [44.0]	33.0 [52.0]

* Abbreviations: Trait-Robustness of Self-Confidence Inventory (TROSCI), Sports Climate Questionnaire (SCQ), Competitive Aggressiveness and Anger Scale (CAAS), Standard Deviation (SD), 95th Percentile (95th %).

**Table 3 sports-10-00115-t003:** Statistical summary of impact frequency metrics for each sample *.

Overall Metric	Session Type	Middle School	High School
		Median [95th %]	Median [95th %]
N	Overall	247.0 [547.0]	317.0 [631.0]
	Practice	146.0 [282.0]	169.0 [356.0]
	Competition	92.0 [275.0]	151.0 [318.0]
Impacts/player/session	Overall	7.4 [18.2]	8.8 [19.7]
	Practice	6.1 [13.6]	6.4 [13.9]
	Competition	10.4 [27.5]	15.4 [32.7]

* Abbreviations: 95th Percentile (95th %), Total Number of Impacts (N), Mean Number of Impacts per Player per Session (Impacts/Player/Session).

**Table 4 sports-10-00115-t004:** Statistical summary of impact magnitude metrics for each sample *.

Overall Metric	Session Type	Middle School	High School
		Median [95th %]	Median [95th %]
Median LA	Overall	18.7 [21.8]	19.2 [23.8]
	Practice	19.1 [22.7]	18.5 [22.5]
	Competition	18.3 [20.6]	19.4 [25.8]
LA95	Overall	50.9 [67.4]	47.7 [65.9]
	Practice	51.7 [77.1]	40.1 [61.9]
	Competition	50.8 [62.3]	53.9 [76.2]
Median RA	Overall	886.9 [1053]	994.4 [1133]
	Practice	904.4 [1095]	985.3 [1102]
	Competition	889.2 [1046]	1024 [1225]
RA95	Overall	2330 [3121]	2493 [3495]
	Practice	2475 [3459]	2224 [3124]
	Competition	2217 [3186]	2701 [3828]
RWE	Overall	0.973 [5.575]	1.358 [4.392]
	Practice	0.328 [2.211]	0.327 [1.192]
	Competition	0.296 [3.364]	1.164 [3.477]

* Abbreviations: 95th Percentile (95th %), 95th Percentile Linear Acceleration (LA95), Median Linear Acceleration (Median LA), 95th Percentile Rotational Acceleration (RA95), Median Rotational Acceleration (Median RA), Risk Weighted Exposure (RWE).

**Table 5 sports-10-00115-t005:** Significant relationships for the middle school sample *.

Survey/Metric	*p* (Survey)	*p* (Age)	*p* (BMI)	*p* (Model)	R^2^
TROSCI					
N-competition	0.045	0.756	0.999	0.999	0.999
Impacts/Player/Session-competition	0.045	0.045	0.999	0.153	0.508
Median RA-practice	0.197	0.283	0.006	0.024	0.720
SCQ					
Median LA-practice	0.025	0.736	0.027	0.042	0.670
Median RA-overall	0.281	0.149	0.003	0.007	0.851
Median RA-practice	0.078	0.370	0.016	0.057	0.691
RWE-competition	0.897	0.034	0.376	0.117	0.547

* Abbreviations: *p*-value (*p*), Trait-Robustness of Self-Confidence Inventory (TROSCI), Sports Climate Questionnaire (SCQ), Total Number of Impacts (N), Mean Number of Impacts per Player per Session (Impacts/Player/Session), Risk Weighted Exposure (RWE), Median Linear Acceleration (Median LA), Median Rotational Acceleration (Median RA).

**Table 6 sports-10-00115-t006:** Significant relationships for the high school sample *.

Survey/Metric	*p* (Survey)	*p* (Age)	*p* (BMI)	*p* (Model)	R^2^
TROSCI					
N-overall	0.434	0.019	0.217	0.045	0.426
N-practice	0.267	0.037	0.079	0.041	0.436
N-competition	0.893	0.041	0.313	0.112	0.321
Impacts/Player/Session-practice	0.450	0.490	0.019	0.044	0.452
Impacts/Player/Session-competition	0.810	0.025	0.571	0.046	0.403
LA95-overall	0.212	0.611	0.018	0.059	0.363
LA95-competition	0.091	0.911	0.034	0.053	0.392
RA95-overall	0.222	0.409	0.002	0.013	0.480
RA95-practice	0.971	0.666	0.005	0.037	0.403
RA95-competition	0.207	0.803	0.013	0.072	0.383
RWE-practice	0.738	0.904	0.036	0.142	0.314
SCQ					
N-overall	0.421	0.019	0.505	0.045	0.428
N-competition	0.376	0.021	0.613	0.042	0.393
Impacts/Player/Session-competition	0.684	0.024	0.723	0.044	0.408
LA95-overall	0.909	0.278	0.024	0.097	0.335
LA95-competition	0.179	0.448	0.018	0.087	0.346
RA95-overall	0.635	0.046	0.003	0.016	0.466
RA95-practice	0.514	0.456	0.016	0.032	0.433
RA95-competition	0.707	0.068	0.046	0.137	0.285
Median RA-overall	0.579	0.023	0.176	0.070	0.349
RWE-overall	0.314	0.062	0.035	0.150	0.291
RWE-practice	0.322	0.743	0.025	0.121	0.351
CAAS					
N-overall	0.483	0.028	0.384	0.048	0.421
N-competition	0.146	0.015	0.666	0.024	0.436
Impacts/Player/Session-competition	0.255	0.017	0.698	0.026	0.452
LA95-overall	0.980	0.521	0.027	0.123	0.296
RA95-overall	0.940	0.262	0.008	0.027	0.428
RA95-practice	0.234	0.464	0.016	0.026	0.413
Median RA-overall	0.175	0.030	0.036	0.044	0.407

*Abbreviations: *p*-value (*p*), Trait-Robustness of Self-Confidence Inventory (TROSCI), Sports Climate Questionnaire (SCQ), Competitive Aggressiveness and Anger Scale (CAAS), Total Number of Impacts (N), Mean Number of Impacts per Player per Session (Impacts/Player/Session), Median Rotational Acceleration (Median RA).

## Data Availability

The biomechanical data from this study will be available via FITBIR.

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
