# Peer review of "Relationship between Aggressiveness, Self-Confidence, and Perceived Coach Support and Head Impact Exposure in Youth Football"

_sports, 2022, doi:10.3390/sports10080115_

Round 1
Reviewer 1 Report
The authors present a unique analysis relating head impact exposure in American football to intrinsic factors reasonably expected to relate to head impact exposure (aggression, self-confidence, and perception of coaching support). This work builds on existing literature that has shown that a not-insignificant component of variation in head impact exposure is due to individual differences.
The authors would be well-served to include citations to previous research investigating some of these factors, such as:
Gellner, R. A., Campolettano, E. T., Smith, E. P., & Rowson, S. (2019). Are specific players more likely to be involved in high-magnitude head impacts in youth football?. Journal of Neurosurgery: Pediatrics, 24(1), 47-53.
_____
For Table 1, it would be helpful to include what the maximum value for each survey is for context, either in the title or in the text that introduces the surveys.
In Table 3, for high school athletes, the RWE data is not complete. Missing the 95%-ile value for competition and the columns are swapped for the Overall header.
In the results section, reference to player age being significantly/not significantly associated with various biomechanical metrics is presented. However, at no point in the manuscript is a summary of player ages presented. The authors could reasonably add in some descriptive stats regarding age of the two populations in order to provide some additional context to the presented results (i.e. do older players have more head impacts because they get more reps in practices and more playing time in competition?)
For middle school athletes, could the higher self-confidence and relation to HIE be related to athletes who have already gone through puberty? How do physical factors (height and weight) for those athletes with high TROSCI scores compare to those with lower scores? How do they relate to measures of HIE? While not the focus of this manuscript, it offers the potential to investigate the combined effect that is likely at play for these vulnerable athletes.
For perceived coaching support, what about the potential that middle school coaches may be parents of children on the team? While this may still occur at the high school level, it is usually less common.
Author Response
Please see the attachment - listed as Reviewer 1

Reviewer 2 Report
Review for MDPI Sports
Title: Relationship Between Aggressiveness, Self-Confidence, and 2 Perceived Coach Support and Head Impact Exposure in Youth 3 Football
General Comments:
· This article is a well-written summary of a study showing contact sport athletes at multiple levels may have their head impact exposure (HIE) influenced by self-confidence, coaching, aggressiveness, and age.
· The general direction of this research is in the correct direction for this field, and I believe the authors’ hypotheses that these factors influence HIE are likely correct.
· The present study is limited by sample size, especially when dividing into smaller groups, as the authors did here.
· The authors failed to reference a few key prior articles. Please review the below articles and consider citing them. There may be other similar references that could enhance the paper’s background strength, as well.
o Drill-Specific Head Impact Exposure in Youth Football Practice – Campolettano et al
§ Consider including on page 1, line 45
o Are Specific Players More Likely to Be Involved In High-Magnitude Head Impacts In Youth Football? – Gellner et al
§ Consider including on page 2, lines 49-50
Methods Comments:
· The authors used three different survey instruments to understand the key factors about participants. Were these surveys designed for youth populations? If not, to what extent do the authors think the wording of the surveys could have affected the results?
· The two teams were compared and contrasted extensively throughout the paper, yet the surveys were given at different time points during the season. This is a significant limitation when comparing the two teams and should be discussed more in the limitations section. I am especially concerned about time- and recall-dependent information, such as the coaching style. The memory of how encouraging a coach was could easily change from mid-season to post-season, as some experiences may not have been had yet by mid-season; alternatively, players may not be able to remember as much by the end of a season.
Statistical Analysis Comments:
· Many factors were related to many biomechanical measures. If the authors were to condense to the few most important biomechanical measures (for example, not separating by session type), would the results still show a relation?
· Age is used as a covariate in each part of the analysis. What is the purpose of this? Is it to be used as a baseline, showing which factor is more/less associated with HIE than age?
o Is age within a team meaningful in this context? For example, a 16-year-old player could be a sophomore or a junior in high school. Could the grade (# years of experience) influence play-time and self-confidence more than age in this situation?
o Also, could age affect the survey results (e.g. greater self-confidence with age, higher trust in a coach with more experience under his/her coaching). If so, how would this affect your results?
Logistical/Aesthetic Comments:
· The tables are difficult to read in their present state. Please reformat so that the delineation between each Overall Metric or Survey is more clearly defined.
o Example of suggestions: https://towardsdatascience.com/five-rules-for-designing-tables-a953a16e50f3
· I also suggest choosing either Mean +/- SD or Median [95th %] to include in the tables. Both describe the dataset, and having both measures for each time in most tables makes for redundant and difficult-to-read tables. Median [95th %] typically describes the right-skewed data from HIE measurements better.
· Figure descriptions seem to be missing (b). This makes it difficult to determine where one description ends and the next begins.
Discussion/Conclusion Comments:
· The description of the HIT System error is vague (lines 313-314). Please see Drill-Specific Head Impact Exposure in Youth Football Practice – Campolettano et al for an example of a more precise description of the HIT System and its associated error.
· Were the survey results correlated to one another? For example, were the players with high self-confidence also more aggressive?

Author Response
Please see the attachment - listed as Reviewer 2

Reviewer 3 Report
This is a well-written and clear manuscript describing research on individual predictors of head impact exposure. The study was conducted in a small number of middle and high school student athletes. Authors found differences between middle and high school samples in exposures occurring during practice versus competitive play, such that middle school athletes maintained more exposure for some metrics during practice, while high school athletes experienced more exposures during competition.Self-esteem, perceived coach support, and anger/aggressiveness also were differentially associated with level of play, and practice vs. competition.
One of the more striking findings of this study is the dissimilarity between practice and competitive associations. Whether this is causally related as offered by the authors, such that those who throw themselves more into practice are better prepared during competition, remains unclear.
Major points:
· Given the number of analyses shown in this paper, I would argue that multiple comparisons correction is warranted, or at least should be commented upon, since many of these marginal results would disappear. The tables with modeling results should show parameter estimates, 95% confidence intervals and p-values, rather than just p-values.
· I would like to see a more traditional table 1 showing the age distributions within each of the samples. I also believe that race should be remarked upon: both the race of the athletes, and the differences by outcome scores. I would expect there to be differences in self-esteem, subjective support from coaches and differences in anger/aggression. If present, such differences would likely be tied to bio-psycho-social factors (not biological), such as via discrimination by coaches, adverse childhood exposures and societal factors that increase self esteem in white but not black children. If there is no variation in the racial makeup of the cohorts, this should be commented upon.
· It would be interesting to see how some of these trends occur over increasing age when the two cohorts are combined: whether there is a sharp change due to coaching approaches upon entrance into high school, or whether the trends are gradual and reflect the maturing of young athletes.
Minor points:
· Legends for a number of tables are missing the “b)” designation for the right hand panel.
· The Abstract would be strengthened with an inclusion of the directions of the most important associations (e.g., “significantly positively associated”) identified by the authors’ analyses.
Author Response
Please see the attachment - listed as Reviewer 3

Round 2
Reviewer 1 Report
The authors did well in responding to reviewers' comments and improving the manuscript.